# Healthy Aging in a Religious Congregation: A Study About Lifestyles and Health Behaviors Among Consecrated Women in Poland and Other Countries

**DOI:** 10.3390/healthcare13080882

**Published:** 2025-04-11

**Authors:** Paulina Teodorczyk, Paweł Najechalski, Maciej Walędziak, Anna Różańska-Walędziak

**Affiliations:** 1Clinical Nursing Department, Faculty of Medicine, Collegium Medicum, Cardinal Stefan Wyszynski University in Warsaw, 01-938 Warsaw, Poland; s.stellacsfn@gmail.com; 2Faculty of Medicine, Lazarski University, 02-662 Warsaw, Poland; pawel.najechalski@lazarski.pl; 3Department of General, Oncological, Metabolic and Thoracic Surgery, Military Institute of Medicine, Szaserów 128 St., 04-141 Warsaw, Poland; 4Department of Human Physiology and Pathophysiology, Faculty of Medicine, Collegium Medicum, Cardinal Stefan Wyszynski University in Warsaw, 01-938 Warsaw, Poland; aniaroza@tlen.pl

**Keywords:** religious, healthy aging, lifestyle, consecrated person, health attitudes

## Abstract

**Background:** The relationship between religiosity and health has been a topic of interest to researchers for many years, especially in the context of the potential positive impact of religion on individual health. A significant correlation between religious affiliation and well-being has been observed among individuals residing in religious communities. The shared lifestyle and values enable analysis of the impact of spirituality on health. **Methods:** The present study aimed to examine the lifestyle of consecrated persons in relation to variables such as nationality, country of ministry, age, and length of time in the community. Furthermore, it sought to ascertain the extent to which community members self-assess their well-being within the context of community life and identify potential areas that require support. **Results:** The survey was conducted using an anonymous questionnaire, which was available in electronic (n = 443) and paper (n = 20) format, depending on the participants’ locations in Poland and abroad. A total of 463 religious women representing 22 different nationalities and serving in 34 countries participated in the survey. The respondents frequently said a balanced diet, rest, and activity are important for good health. On average, 57% of the participants (n = 264) said their lifestyle was healthy, with the most responses coming from women aged 65 and over (73%). This could be due to a positive attitude towards healthy behaviors, regular exercise, better stress coping, and lower depression symptoms. These differences were statistically significant (*p* < 0.05). **Conclusions:** Participants of non-Polish nationality who live outside Poland perceived their lifestyle as healthy significantly more often than Polish nationals and those living in Poland. The study shows that religiosity and spirituality may improve subjective health assessments, especially among the elderly, which could be the basis for research on wider populations.

## 1. Introduction

The relationship between health and religion has been a topic of interest to researchers since the beginning of recorded history. A review of the scientific literature from the last century reveals a significant body of evidence indicating that religious practices have, across cultures, a beneficial effect on health outcomes [1,2,3,4,5,6]. It is, nevertheless, occasionally observed that this is not a straightforward issue, given that those who identify as atheists represent a relatively limited sample, which does not permit a definitive conclusion to be drawn regarding the influence of religion on health. This observation emphasizes the necessity for further research [7].

In the course of research conducted on the mechanisms through which religion exerts an influence on health, it has been observed that the references in question can be classified as either positive or negative. Positive religious coping has been defined as the utilization of a relationship with God through prayer and transcendence, and studies have indicated that individuals who employ this coping mechanism exhibit higher levels of mental health in comparison to those who engage in negative religious coping, which is characterized by the attribution of blame to God for personal difficulties [8,9]. The impact of gratitude was also emphasized, with evidence indicating that its expression is frequently observed in spiritual practices such as prayer and meditation. These practices have been shown to enhance psychological well-being and improve self-perceived health. Individuals who regularly engage in gratitude practices have been observed to experience more frequent positive emotions, adopt adaptive coping strategies, and demonstrate a heightened sense of life satisfaction [10,11].

Furthermore, it is imperative to acknowledge the pivotal role of the immediate environment, encompassing the network of social and psychological connections, in shaping health attitudes. This environment exerts a significant influence on an individual’s health-related attitudes and behaviors. This assertion is particularly illustrated in the theory of planned behavior, which posits that behavior is guided by three main types of factors: behavioral, normative, and control beliefs. The first of these factors pertains to the individual’s personal conviction and evaluation of the phenomenon under consideration. The second is associated with the behavior of social norms, and the third is related to the capability and satisfaction derived from the exercise of agency. The model can indicate whether group identification, for example, as a member of a religious community, has a direct impact on health behavior [12]. The formation of health attitudes is of great consequence in the context of population aging and demographic change, given that these factors affect the global population [13,14].

In the context of this study, a particular emphasis is placed on the relationship between religion and spirituality and health and aging, with a specific focus on a unique group that has been found to be of particular interest in this area: religious nuns. A seminal study by Snowdon (n = 678) investigated the relationship between lifestyle and the development of Alzheimer’s disease in Catholic nuns, finding that the former was associated with a lower risk of developing the disease and a longer life expectancy. Higher education, higher levels of linguistic ability, and a positive outlook on the future were neuroprotective [15]. It has furthermore been posited that religious practices may foster qualities such as optimism, a propensity to forgive, and a sense of gratitude, which have been identified as markers of successful aging [16]. These aspects confirm that aging cannot be viewed solely in terms of biomedical theories but is also influenced by subjective elements [17]. It has also been found that spirituality, which enhances mental well-being and provides a foundation for finding meaning in life, helps to cope with the challenges of aging and is considered one of the five main categories in the concept of successful aging, along with social well-being, mental well-being, physical health, and environmental and economic security [18,19].

The daily routine of religious women shows significant similarities in terms of sleep patterns, working hours, physical and mental activity, and rest [20]. This stability of lifestyle is ambiguous, mainly because of the country of service. In our study, we took this into account by creating three groups of Sisters according to country of service: Poland, mission countries, and non-mission countries. This is important in the context of understanding the impact of the community on health in the context of different environmental and cultural conditions. The opportunity to conduct the study in an international group including people of various ages offers valuable insights into the factors influencing the aging process, particularly in people who have dedicated their lives to a religious vocation and report high levels of life satisfaction and adherence to a healthy lifestyle.

The principal objective of this study was to examine the lifestyle of consecrated persons in relation to a number of factors, including nationality, country of ministry, age, and length of stay in the community.

Another objective of the study was to ascertain how members of religious communities evaluate their well-being in relation to community life and to identify potential avenues for improvement.

## 2. Materials and Methods

This study was designed as an anonymous survey with the objective of determining how members of religious communities assess their well-being in relation to community life and to identify potential avenues for improvement. The survey was disseminated via email in an electronic version (443) and in a paper version (20), depending on the participants’ place of residence: Poland versus countries other than Poland.

Due to the diversity of the sample, the data were divided into three groups: missionary countries, non-missionary countries, and Poland, according to the Congregation for the Evangelization of Peoples. This division is predicated on the endeavor to enhance comprehension of the manner in which external conditions influence the lifestyle of religious communities. According to the Congregation for the Evangelization of Peoples, three criteria are taken into account when determining whether a country or territory is considered mission territory. Firstly, a small number of Catholics and the associated need for first evangelization. Secondly, a lack of sufficient local vocations and priests to ensure the proper functioning of the diocese. Thirdly, a lack of finances even to meet basic needs. This has a direct impact on the nature of religious communities, which are smaller, not engaged in profit-oriented activities but in charity, and usually have a lower socio-economic status compared to non-mission countries [21]. In accordance with these stipulated guidelines, the non-missionary countries included the USA and countries in Europe, Israel, and Australia. The missionary countries involved countries from Asia, Eastern Europe, Africa, and South America.

The questionnaire employed in this study is an original instrument developed by the research team. This instrument was developed by the research team, who had extensive clinical experience and had conducted a comprehensive review of the relevant literature. To ensure the validity and reliability of the instrument, a pilot study was conducted to assess the questions’ clarity, unambiguity, and content adequacy. Following the analysis of the pilot study results, the research team made the necessary modifications to the questionnaire, thus ensuring its validation prior to its implementation in the main study

The initial section of the survey encompassed fundamental data regarding age, nationality, educational background, country of ministry, and the duration of service within the congregation. The subsequent section delved into dietary habits, rest and exercise routines, the occurrence of chronic illnesses, and the administration of medication, including sleeping tablets. The concluding section of the questionnaire pertained to beliefs and attitudes towards health, along with the impact of specific elements of life within the religious community on the formation of these attitudes and the general state of health.

Inclusion Criteria:Members of female apostolic religious communities in the Catholic ChurchOver 18 years of ageParticipants must belong to congregations that are international and established under papal lawParticipants must wear religious garb

Exclusion Criteria:Refusal to join research.

The questionnaire was distributed in consultation with the Conference of Major Superiors of Religious Congregations in Poland.

### 2.1. Statistical Analysis

The relationships between the unmeasured characteristics were examined using the chi-square test of independence in a multivariable (contingency) table (χ^2^). A statistical test was also used to obtain a correlation result, in which the Spearman’s rank correlation coefficient (rho) was calculated. This coefficient is used to determine the strength and direction of the monotonic relationship between variables. Values approaching 1 or −1 are indicative of a strong relationship (positive or negative, respectively), while values close to 0 indicate a weak or non-existent relationship. This study assumed a significance level of α = 0.05 and a confidence level of 95%. The relationship between age and length of stay in the community was determined by calculating the mean, median, and standard deviation. The normality of the age distribution was confirmed by Lilliefors and Shapiro–Wilk tests. The sample size corresponding to a *p*-value of less than 0.05 and a confidence level of 95% for the number of nuns worldwide (608,958 in 2023) was 384.

### 2.2. Ethical Considerations

This study was anonymous and performed in accordance with the ethical standards laid out in the 1964 Declaration of Helsinki and its later amendments (Fortaleza). Participants were informed about the aim of this study, and informed consent was obtained from every participant. The approval from the Bioethics Committee of the National Medical Institute of the Ministry of the Interior and Administration in Warsaw, Poland, with code 25/2024, was obtained on 19 April 2024.

## 3. Results

### 3.1. Participant Demographic Characteristics

The research involved 463 religious women of 22 nationalities, working in 34 countries. The age range was 22 to 91, with a median of 50 (SD ± 14.1) and a mean age of 51.4 (IRQ 21). To ensure the quality of the results, the participants were divided into three age-related groups: 18–49, 50–64, and 65+. The mean number of years in the congregation was 30.6 (SD ± 15.8). Three groups were identified based on the mean years in the congregation: under 10 years, between 11 and 24 years, and above 25 years. It is customary for a Sister to become a full member of the congregation by taking final vows after approximately ten years of service. The first significant jubilee, which occurs after twenty-five years of service, often presents an opportunity for reflection and evaluation. The majority of participants had received a university education, including some participants who had received medical education or had less than a university education. These data are presented in Table 1.

The majority of participants in this study who were over 65 years of age originated from non-missionary countries, representing 70% of the total sample. In contrast, the largest age group of Sisters, comprising one in three of the participants, was from Poland. Conversely, in the youngest age group, between 18 and 50, the majority, representing 58% of the total sample, are from Poland. Further detailed data are presented in Table 2.

### 3.2. Physical Activity

A significant proportion of non-Polish participants (approximately 40%) engaged in regular physical activity during their leisure time. This figure was approximately twice as high as that observed among Polish consecrated women (17%). A similar correlation was observed in data summaries from various countries of ministry, with approximately 30% of individuals residing outside of Poland having reported daily physical activity, including 36% of those working in mission countries, *p* < 0.05. The correlation between physical activity during free time and age was a significant statistical factor, with a twofold increase observed in the oldest age group (40%), compared to the youngest age group (20%). This finding was corroborated by a Spearman test, which demonstrated a positive correlation between age and physical activity with a significance level of less than 0.05. Further details are presented in Table 3.

The prevalence of self-reported physical activity during professional work was higher among individuals living in missionary countries (57%) and among those in the youngest age group (49%), as illustrated in Table 4.

Furthermore, participants between 50 and 64 years old demonstrated the highest prevalence of daily sedentary postures during working hours, with a rate of 59% compared to 40% in the youngest age group and 55% in the oldest, *p* < 0.05.

### 3.3. Food and Beverage Consumption

The majority of participants (over 80%) reported consuming regular meals on a daily basis, as well as maintaining an adequate hydration level (over 70%). These findings were not influenced by factors such as nationality, age, country of ministry, or length of service within the congregation. The aforementioned data are presented in Table 5.

A total of 119 participants (25%) reported that they consumed animal fats and highly processed foods on a daily basis. Additionally, 132 participants (28%) indicated that they consumed sweets and/or sweetened beverages at the same frequency. It must be noted that the lack of a statistically significant result does not permit the drawing of any definitive conclusions. However, the observed trend suggests that in missionary countries, consecrated women tend to consume the lowest quantities of highly processed products and sweetened beverages, 13% and 11% of participants, with, respectively, 30% and 31% in Poland and 24% and 32% in non-missionary countries.

Furthermore, a notable correlation was observed between the frequency of daily consumption of sweets and adding salt to meals on a regular basis and the number of years spent in the congregation. The highest number of participants who admitted regularly adding salt to their meals, 39% (n = 16; *p* < 0.05), and daily consumption of sweets, 46% (n = 19, *p* < 0.05), was observed in the group who had been in the congregation for less than 10 years. The lowest level of sweet consumption was observed among participants who had been in the community for more than 25 years—25% (n = 68). The prevalence of the habit of adding salt to meals was observed among individuals who had spent from 11 to 24 years in the community—22% (n = 36).

The highest consumption of fruits and vegetables was observed among participants residing in non-missionary countries, with a prevalence of 57% (n = 87; *p*-value < 0.05), compared to 33% in Poland and 54% in missionary countries. The highest number of consecrated women who practiced fasting was present in the middle age group (39%) and in missionary countries (44%). One in three participants aged over 65 reported that they never ate in order to improve their mood, compared to only one in six participants in the youngest age group. The detailed data are presented in Table 5.

The highest prevalence of daily coffee consumption was observed among Polish nuns (82%, n = 260). A statistically significant correlation was observed between coffee consumption and both the number of years spent in the congregation and the age at which one joined the congregation. The highest rates of coffee consumption were found in the group of participants who had lived in the community for more than 25 years (81%, n = 218) and those who had joined the community before the age of 25 (78%, n = 296).

### 3.4. Psychological Distress and Chronic Conditions

A statistically significant correlation was observed between psychological distress and age. The lowest rates of low mood and stress were observed in the oldest group of consecrated persons. Specifically, 21% (n = 17) of the participants reported having never experienced low mood, while 39% (n = 32) indicated that they never experienced stress. Additionally, the lowest proportion of individuals who indicated having excessive workload on a daily basis (9%, n = 7) was also found in the oldest group, as well as the highest proportion of participants who felt they had sufficient time for rest (34%, n = 28). The Spearman test confirmed a significant negative correlation between age and the experience of low mood (*p* < 0.05, rho = −0.164), stress problems (*p* < 0.05, rho = −0.284), and the use of food to improve mood (*p* < 0.05, rho = −0.201). These results indicate that the lowest levels of intensity of these phenomena are observed in the oldest age group. Nearly half of the participants aged 65 and above reported having satisfactory sleep. Further details are presented in Table 6.

Moreover, the findings of the chi-square test substantiated the existence of statistically significant correlations. These correlations indicate that individuals who most frequently report experiencing stress also report low mood (strong positive correlation rho = 0.641), anxiety (strong positive correlation rho = 0.637), a feeling of not having enough time to rest (strong positive correlation rho = 0.614), and a feeling of being overworked (strong positive correlation rho = 0.519). In addition, a moderate positive correlation (rho = 0.331) was observed between depressed mood and the consumption of food for the purpose of mood enhancement. It is noteworthy that the *p*-value for all the aforementioned correlations was less than 0.05, thereby maintaining a significance level of α = 0.05 and a confidence level of 95%.

Of the 463 individuals who participated in the study, 207 were diagnosed with one or more chronic conditions, with a notable increase in their prevalence positively correlated with age (*p* < 0.05), with rates twofold higher in the older age group when compared to the younger group (63% vs. 30%). The rate of participants who reported having chronic conditions was three times higher in Poland (49%) than in missionary countries. A comparable pattern was identified in relation to the consumption of medications prescribed by medical professionals. However, the observed rates were 20% higher in each age group. Detailed data are presented in Figure 1. The mean prevalence of sleep medication use was 12.3% (n = 57), with no significant correlation with age, nationality, or country of service.

### 3.5. Health Attitudes

A majority of individuals residing in religious communities outside Poland (over 50%) indicated that a healthy diet was a priority for them, they wanted to maintain a balance between rest and work, they engaged in regular physical activity, and they prioritized adequate sleep. Additionally, 29% of Polish participants indicated that a balanced diet, regular meals, and physical activity were of high importance to them. However, they did not prioritize sufficient rest and sleep.

The relationship between health attitudes and nationality is presented in Figure 2.

A notable correlation was also identified in relation to the description of health attitudes and age. Those aged 65 and over tended to have the most positive health attitudes, while those aged between 50 and 64 tended to have the least positive attitudes. The participants were invited to indicate their health attitudes from a list of five—the first being the most correct, the fifth being the most incorrect—and to select the one that best described their daily experience. A comprehensive description of the attitudes and a more detailed presentation of the data can be found in Figure 3.

Almost all participants declared that nutrition (n = 454), physical activity (n = 461), and rest (n = 463) had an essential impact on maintaining good health. A total of 57% (n = 264) of the sample of Sisters indicated that they adhere to a healthy lifestyle. Participants over 65 years old were most likely to have stated that they led a healthy lifestyle, with a prevalence of 73%, which is consistent with the aforementioned results about different daily habits, *p* < 0.05. Participants of non-Polish nationalities and living outside of Poland significantly more often considered their lifestyle as healthy than those of Polish nationality or living in Poland, *p* < 0.001. These results are presented in Figure 4.

Furthermore, a highly statistically significant correlation (*p* < 0.001) was identified between the declaration of healthy lifestyles and health attitudes. Almost half of those residing in a religious community indicate that they possess the most positive attitudes across all areas of life, while only 3% of them asserting that they have the least favorable attitudes.

A strong correlation was confirmed between chronic illness and healthy attitudes. Over one-third of the Sisters who had not suffered from chronic illness exhibited the most positive attitudes related to health, while only 6% of this group displayed the least positive attitudes. The aforementioned data are illustrated in Figure 5.

This study also demonstrated, employing a *t*-student test, a statistically significant relationship between the statement ‘I lead a healthy lifestyle’ and the statement ‘Community has a positive impact on health’. Both statements were treated as single elements of self-assessment. Amongst those who believe that community has a positive impact on health, 69.1% assert that they lead a healthy lifestyle, 10.7% claim that they do not lead a healthy lifestyle, and 20.2% express difficulty in making a clear assessment of their lifestyle. Conversely, among those who do not subscribe to the belief that community life exerts a positive influence on health, only 26.3% affirm a healthy lifestyle, 39.5% do not, and 34.2% find it challenging to make a definitive assessment. A statistically significant relationship was also identified between individuals who perceive community life positively and negatively in relation to a healthy lifestyle (t = 5.21; *p* < 0.05). These results suggest a strong positive relationship: people who perceive community life as beneficial to their health are much more likely to report having a healthy lifestyle. This finding may imply that the community plays a significant role in fostering motivation or providing support, thereby facilitating the maintenance of healthy habits.

## 4. Discussion

The results of numerous studies conducted over the past century indicate a notable correlation between spiritual practices and physical health. Some studies additionally indicate that this relationship becomes more pronounced as individuals grow older [6,22,23]. Our investigation focused on a cohort of nuns who had embraced a way of life centered upon a profound engagement with catholic spirituality, manifesting in their engagement with religious practices. Research by Henderson et al. among college students in the USA suggests that regular religious practice can improve life satisfaction and well-being [24]. The findings were comparable to those from our study, which revealed that over half (57%) of participants whose lives were centered around religious practices also indicated that they followed a healthy lifestyle. A mere 16% of the participants reported that their lifestyle was unsatisfactory. This finding aligns with the Theory of Planned Behavior, a concept predicated on the notion that normative beliefs exert a significant influence on the shaping of behavior. The findings of our study demonstrate a robust positive correlation, thereby suggesting that individuals who perceive community life as being conducive to health are significantly more inclined to assert that they adhere to a healthy lifestyle. This finding suggests the potential of community as a significant motivating or supporting factor in the maintenance of healthy habits [12].

Our findings additionally indicate that the oldest age group exhibited the highest prevalence of positive perception of their health, with an impressive 73% of the cohort comprising women over 65 years of age. This result is significantly higher than that observed in other age groups and can be attributed to a number of factors. Firstly, it has been demonstrated that the most positive description of health behaviors was the one most frequently chosen in this age group. Furthermore, the assessment of individual lifestyle elements, including physical exercise, stress management techniques, and infrequent bouts of bad mood, has been demonstrated to have a positive impact on lifestyle self-assessment. This phenomenon persists despite the widespread presence of chronic diseases and the resulting limitations. This phenomenon may be attributed to the accumulation of life experience, which is often accompanied by increased resilience [25]. This is particularly pertinent in light of the findings that a healthy lifestyle exerts a comparable effect on alleviating symptoms associated with the aging process as genetic factors [26]. A comparison of our results with the results of studies conducted in groups of adherents of other religions reveals that they are consistent with ours. A study by Min Tan et al. indicates that in the Muslim community, older adult men (i.e., those over 55 years of age) exhibit higher levels of well-being that are positively correlated with the level of religiosity [27].

The findings of Eilat Alat and colleagues demonstrate that elevated levels of well-being and enhanced health outcomes are associated with religiosity, a phenomenon that is particularly evident among ultra-Orthodox Jewish men [28]. A comparison of the dietary habits of consecrated women and ultra-Orthodox Jewish women in Israel revealed that the former group consumed fruits and vegetables more frequently, with 48% of them eating them daily, compared to only 12.7% of the latter group. The consumption of sweets and/or sweetened beverages on a daily basis was observed among nearly one-third of the consecrated women participants. In contrast, a study by Leiter et al. indicated that 18.9% of the ultra-Orthodox Jewish women consumed more than five cups of sweetened beverages per week. Furthermore, it was observed that nearly two-thirds of the participants slept for less than seven hours per day. In contrast, our research in a non-missionary country, such as Israel, indicated that nearly 39% of the participants reported having adequate sleep [29].

Engaging in physical activity has been demonstrated to preserve cognitive and physical functions in the elderly [30]. Furthermore, it has been shown to contribute to healthy aging in both adult and elderly populations and the prevention of diseases typical in old age [31,32,33]. This finding is corroborated by our study, which revealed that the proportion of participants who self-reported leading a healthy lifestyle was notably high among those aged 65 and above (73%). Additionally, this age group demonstrated the highest prevalence of daily physical activity—40% of participants—which, in addition to adequate hydration, nutrition, life philosophy, spirituality, and social support, is considered a factor in healthy aging. The James et al. study addressed the importance of regular physical activity in leisure time for the development of neurodegenerative disorders. The results of this study suggested that individuals who engage in regular physical activity exhibit enhanced cognitive performance and an enlarged hippocampus, even among those diagnosed with early-stage Alzheimer’s disease. A notable finding was the observation that women exhibited heightened sensitivity to physical inactivity, resulting in accelerated cognitive decline [34].

Research into the use of public health services in Poland showed that 69% of participants drank enough water, and two out of three ate regularly. In our research, the proportion of participants drinking an adequate amount of water was comparable to the general Polish population, but there was a higher proportion of Sisters who had regular meals—over 80%—when compared to the general Polish population. The prevalence of regular physical activity was 41%, which is higher than the rate observed in our study, in which 17% of Sisters residing in Poland indicated engaging in physical activity on a daily basis, and 41% of Sisters reported doing so at least once a week. According to the National Health Fund, 67% of Polish people get enough sleep and 63% feel they can cope with stress, with the highest percentage found in the oldest age group—over 60–80% and 73%, respectively. The same trend can be observed in our research, but the results were less satisfactory, as only 38% of all participants reported having had adequate sleep, and, respectively, 49% in the group over 65 years [31].

A comparable study conducted in a similar setting of German–Austrian monasteries revealed that 33.1% of the nuns and monastics surveyed engaged in relaxation practices at least once a week, while an additional 36.8% incorporated relaxation and meditation exercises into their daily routine. Moreover, 54.7% of the participants rated their health as excellent, while 20.8% reported frequently having had feelings of nervousness or stress [33,35]. In contrast, a study conducted among Catholic priests in Latin America revealed that three out of five experienced job burnout. In these studies, psychological stress has been identified as a significant risk factor for the development of professional burnout, particularly in groups of individuals with a substantial workload and a focus on providing assistance to others [36,37]. The research conducted by Vangaikar and colleagues indicates that individuals who adhere to a diverse range of religious beliefs tend to exhibit superior levels of positive mental health compared to those who do not adhere to any particular religion [38]. The results of our survey indicate that approximately 39% of those in the oldest group of consecrated persons had never experienced stress and 22% had never experienced a depressed mood. These findings suggest that living in a religious community may have a positive impact on mental health, particularly in the elderly. A comparable relationship was identified in a study conducted by Stearns and colleagues, where individuals aged 50 and above exhibited higher levels of religiosity, while younger individuals demonstrated a greater proclivity towards reporting depressive symptoms [39]. Similar results were found in a study by Wells et al. conducted on a group of 50 Roman Catholic nuns over the age of 55. The aim was to compare their ability to adapt to changing circumstances in terms of physical fitness, self-rated physical and mental health, and symptoms of depression. Lower levels of depression, higher self-rated mental health, and better physical fitness characterized Sisters with greater adaptability [40]. Contemporary psychological concepts of aging emphasize the role of subjective aspects as well as factors such as physical activity and social interaction. A healthy lifestyle, characterized by a proper diet, adequate hydration, regular exercise, and goal-setting, has been demonstrated to mitigate the limitations associated with advancing age. A growing body of research has demonstrated the positive impact of spirituality on the lives of older individuals, particularly in terms of enhancing resilience and providing a sense of purpose during challenging periods [17,18].

It is also noteworthy that in our study, the oldest participants were the most likely to provide the most positive descriptions of their health attitudes. Almost half of the respondents above 65 years old indicated that maintaining a healthy diet is of great importance to them. They also expressed appreciation for the role of achieving an appropriate balance between rest and work. In addition, they highlighted the benefits of physical activity and sufficient levels of sleep. This positive fact is of great significance from the perspective of public health. Attitudes towards the health of older people can exert a considerable influence on the overall aging of the population and thus warrant thorough investigation [41].

In our study, the prevalence of chronic diseases and prescription drug use increased with age. Research conducted in Korea showed that the optimal use of prescription medicines was necessary for older adults with chronic conditions to achieve good health outcomes and quality of life. Public health policies should be complemented for vulnerable groups, such as older people living alone, including pharmaceutical benefit plans, to optimize the management of their chronic conditions [42]. A study of the relationship between wealth and the use of sleep medication revealed a correlation between higher levels of use and higher economic status, with figures ranging between 15% and 20% among older participants in the National Health and Aging Trends Study [43]. It is important to note, however, that at present, sleep problems are often medicalized, which can prevent individuals from taking on responsibilities, particularly those of an older age [44]. In our study, 12% of respondents reported that they regularly took sleep medications. No significant correlation was found between age, nationality, or country of service. Moreover, our study identified an unparalleled trend whereby the consumption of long-term prescription drugs is, on average, 20% higher than the prevalence of chronic diseases in each age cohort. This may be indicative of a deficiency in disease awareness or may be indicative of a tendency to treat disease symptoms without recognizing the underlying disease process. The etiology may also be the result of existing subclinical conditions or the need to supplement prescription vitamin preparations, such as vitamin D, at higher doses. This observation may be considered surprising and merits further investigation in future studies.

### Limitations of the Study

The initial limitation of our survey is its format, specifically the low availability of paper questionnaires, which would facilitate the participation of older community members.

A further limitation is the absence of questions about substance abuse. However, there was a genuine apprehension that this would result in a reduction in the number of survey participants.

Furthermore, the survey does not inquire about preventive medical examinations. This is a deliberate decision, given the international scope of the survey and the varying, and often constrained, access to medical care, particularly in mission countries.

The possible limitations of our study are recall bias, social desirability bias, and the subjectivity of patients’ opinions. However, there was no incentive to introduce dishonesty into responses. In the current study, no tools specifically designed to control confounding factors were included.

Notwithstanding the international scope of the study, the majority of participants are of Polish nationality, despite serving in a country other than Poland, which may have an impact on the study’s outcomes. This should be considered when interpreting the results in missionary countries where the presence of native nuns is lower. Therefore, the results should be interpreted from the perspective of the health of the missionary nun.

## 5. Conclusions

The results of the present study suggest that religiosity and spirituality have the potential to exert a beneficial influence on the assessment of health status, a phenomenon that appears to increase with age and is particularly evident amongst cohorts of Roman Catholic women residing in religious communities.

The findings of this study indicate a notable difference of nearly 20% between the regular use of medically prescribed drugs and the occurrence of chronic diseases. This observation underscores the necessity for a more in-depth analysis and further research into the quality of medical care, awareness of subclinical symptoms, and the accuracy of chronic disease diagnosis.

A notable association has been identified, suggesting that individuals from the oldest age cohort residing outside of Poland tend to report a healthier lifestyle and more positive health attitudes. This association merits further investigation to ascertain which elements of community life most significantly influence these patterns and the extent to which cultural factors shape these outcomes. The identification of these associations could inform strategies for the promotion of healthier lifestyles within the general population. Furthermore, exploring potential pathways for integrating these insights into Polish religious communities may support the development of similarly favorable health attitudes in the future.

## Figures and Tables

**Figure 1 healthcare-13-00882-f001:**
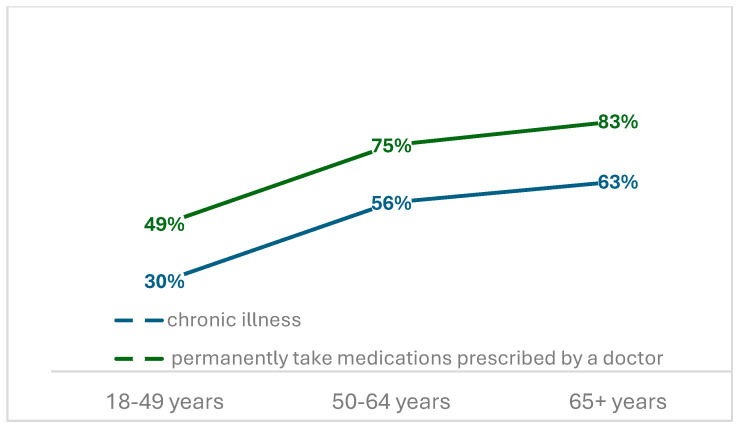
Chronic illness and permanently take medications in reference to age.

**Figure 2 healthcare-13-00882-f002:**
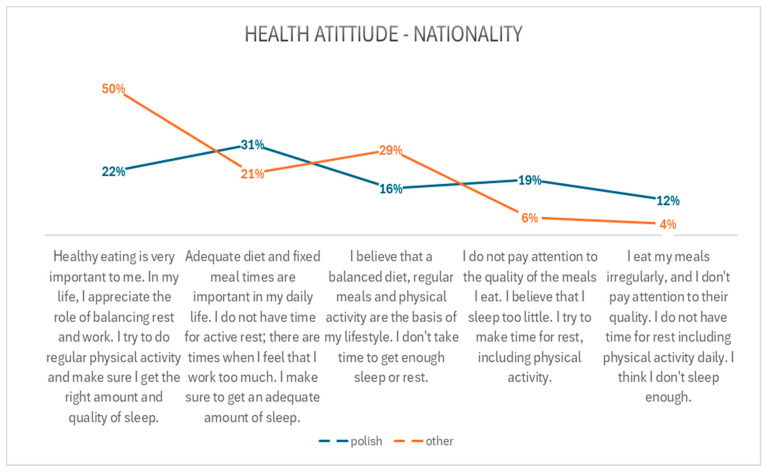
Relationship between health attitudes and nationality.

**Figure 3 healthcare-13-00882-f003:**
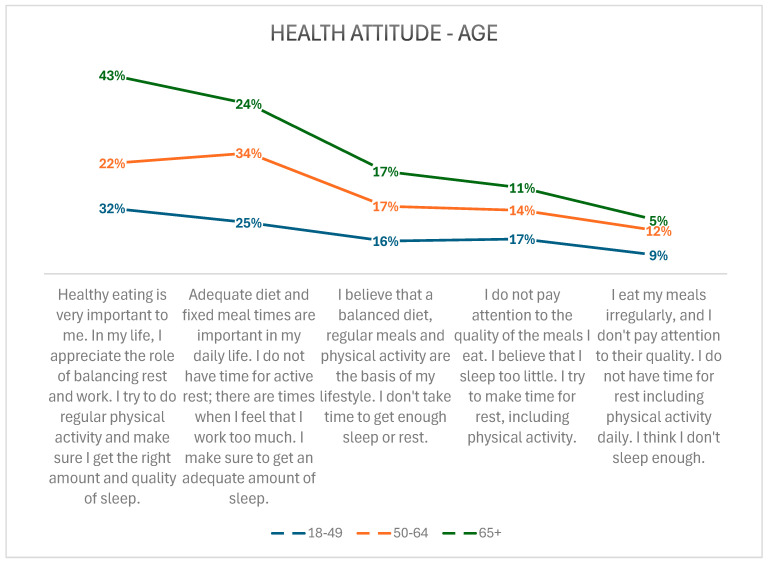
Relationship between health attitudes and age.

**Figure 4 healthcare-13-00882-f004:**
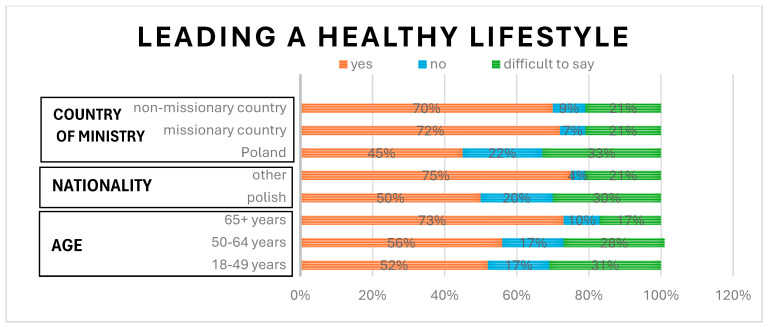
The relationship between nationality, country of ministry, and age and leading a healthy lifestyle.

**Figure 5 healthcare-13-00882-f005:**
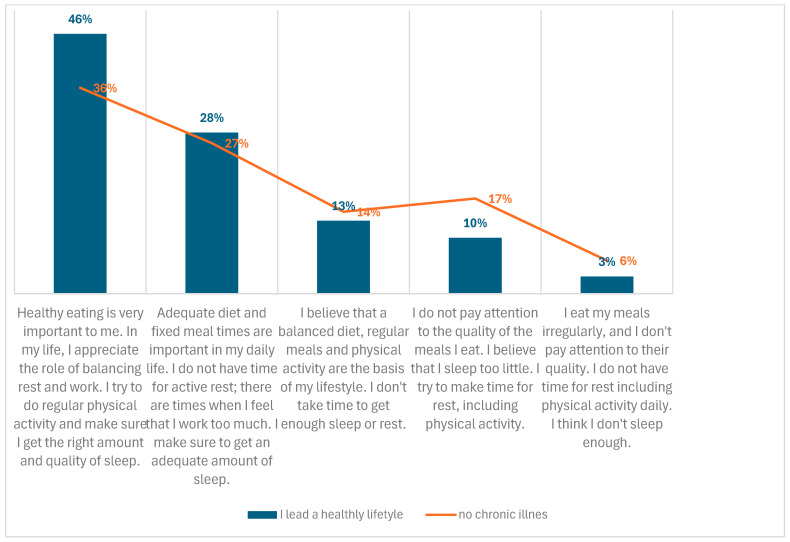
Lead healthy lifestyle, no chronic illness related to attitudes.

**Table 1 healthcare-13-00882-t001:** Participant demographic characteristics.

Median Age, Years (SD)		50 (±14.1)
Mean age, years (IRQ)		51.9 (21)
Age n (%)	18–49	228 (49%)
	50–64	151 (33%)
	65+	81 (18%)
The mean number of years in the congregation (SD)		30.6 (±15.8)
Number of years in the congregation n (%)	0–10	41 (8.8%)
	11–24	133 (28.7%)
	25+	268 (57.9%)
	No answer	21 (4.5%)
Nationality n (%)	Polish	323 (69.8%)
	Other	140 (30.2%)
Country of ministry n (%)	Poland	250 (54%)
	Missionary country	61 (13.2%)
	Non-missionary country	148 (32%)
	No answer	4 (0.8%)
Education n (%)	Medical	20 (4.3%)
	Less than a university education	60 (13%)
	University education	383 (82.7%)

**Table 2 healthcare-13-00882-t002:** Age and country of ministry.

Age	Country of Ministry		
	Poland	Missionary Country	Non-Missionary Country
18–49	132	45	50
%	58.15%	19.82%	22.03%
50–64	94	14	41
%	63.09%	9.40%	27.52%
65+	23	2	57
%	28.05%	2.44%	69.51%

**Table 3 healthcare-13-00882-t003:** Correlation between daily physical activity and nationality, age, and country of ministry.

Physical Activity for at Least 30 min a Day as Part of Leisure Activities	Nationality			Country of Ministry				Age (Years)			
	Polish	Other		Poland	Missionary Country	Non-Missionary Country		18–49	50–64	65+	
Every day	17%	39%	*p* < 0.05	16%	36%	30%	*p* < 0.05	20%	20%	40%	*p* < 0.05
At least once a week	41%	35%		39%	38%	39%		41%	46%	22%	
At least once a month	20%	9%		22%	10%	11%		17%	18%	13%	
Several times a year	16%	9%		16%	11%	12%		15%	13%	12%	
Never	6%	8%		7%	5%	7%		7%	4%	12%	

**Table 4 healthcare-13-00882-t004:** Correlation between daily physical activity at work and nationality, age, and country of service.

Variable	Perform Physical Activity at Least 30 min a Day as Part of Professional Work (%)				
	Everyday	At Least Once a Week	At Least Once a Month	Several Times a Year	Never
Nationality					
Polish	50	22	9	7	13
Other	40	22	6	9	23
*p*-value	=0.05				
Country of ministry					
Poland	48	21	10	8	13
Missionary country	57	28	5	3	7
Non-missionary country	40	21	5	9	25
*p*-value	<0.05				
Age					
18–49	49	22	8	9	11
50–64	46	24	9	5	17
65+	41	18	5	7	28
*p*-value	<0.05				

**Table 5 healthcare-13-00882-t005:** Impact of age and country of service on eating-related health behavior.

	Age (Years)			*p*	Country of Ministry			*p*
	18–49	50–64	65+		Poland	Missionary Country	Non-Missionary Country	
Eat vegetables and fruits at least 3 times a day n (%)	90 (39%)	67 (44%)	43 (52%)	*p* = 0.12	83 (33%)	33 (54%)	85 (57%)	*p* < 0.05
Practice fasting at least once a week	70 (31%)	59 (39%)	20 (24%)	*p* < 0.05	69 (28%)	27 (44%)	52 (35%)	*p* < 0.05
Never eat to improve your mood n (%)	32 (14%)	26 (17%)	26 (32%)	*p* < 0.05	44 (18%)	9 (15%)	31 (21%)	*p* < 0.05
Eat regular meals every day n (%)	189 (83%)	136 (90%)	69 (84%)	*p* = 0.78	220 (88%)	50 (82%)	122 (82%)	*p* = 0.55
Every day drink at least 1.5 L of fluids daily n (%)	155 (68%)	108 (72%)	55 (67%)	*p* = 0.06	175 (70%)	46 (75%)	96 (65%)	*p* = 0.54

**Table 6 healthcare-13-00882-t006:** The relationship between age, country of ministry, rest, and mental well-being.

	Age (Years)			*p*	Country of Ministry			*p*
	18–49	50–64	65+		Poland	Missionary Country	Non-Misionary Country	
Sleep long enough every day n (%)	59 (26%)	53 (35%)	40 (49%)	*p* < 0.05	70 (30%)	20 (33%)	57 (39%)	*p* = 0.64
Every day nap during the day	48 (21%)	33 (22%)	23 (28%)	*p* < 0.05	58 (23%)	18 (30%)	27 (18%)	*p* < 0.05
Have time to rest n (%)	17 (7%)	16 (11%)	28 (34%)	*p* < 0.05	47 (19%)	7 (11%)	14 (9%)	*p* < 0.05
too much work every day n (%)	40 (18%)	28 (19%)	7 (9%)	*p* < 0.05	46 (18%)	11 (18%)	17 (11%)	*p* = 0.10
Never experience low mood n (%)	14 (6%)	11 (7%)	17 (21%)	*p* < 0.05	6%	11%	13%	*p* < 0.05
Never experiences stress n (%)	25 (11%)	22 (15%)	32 (39%)	*p* < 0.05	38 (15%)	8 (13%)	33 (22%)	*p* = 0.07

## Data Availability

The data presented in this study are available on request from the corresponding author due to (specify the reason for the restriction).

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
