# Peer review of "Healthy Aging in a Religious Congregation: A Study About Lifestyles and Health Behaviors Among Consecrated Women in Poland and Other Countries"

_healthcare, 2025, doi:10.3390/healthcare13080882_

Round 1
Reviewer 1 Report
Comments and Suggestions for Authors
This is an interesting study on life styles and health behaviours among consecrated women. I enjoyed reading the paper and appreciate the large sample size of this study. I have several comments to improve the manuscript further:
1. First, the introduction discusses the association between religion and health rather broadly, but it does not clearly articulate a specific theoretical framework. I would suggest the authors to clearly identify a psychological theory or model (e.g., Health Belief Model, Theory of Planned Behavior) to frame the research
2. Similarly, the introduction omits detailed exploration of potential mechanisms by which religiosity influences health. There are several potential mechanisms that can be discussed in the literature review. For instance, gratitude, often cultivated through religious practices like prayer and reflection, has been linked to improved emotional well-being and physical health. Social support within religious communities can buffer stress and reinforce health-promoting behaviors. Religious coping, such as viewing adversity through a spiritual lens, may help individuals manage stress more effectively and maintain psychological resilience. These should be highlighted to strengthen the introduction. Here are some papers that I think will be helpful and should be incorporated:
Social support and health: a review of physiological processes potentially underlying links to disease outcomes. (2006). Journal of behavioral medicine, 29, 377-387.
Dispositional gratitude, health-related factors, and lipid profiles in midlife: a biomarker study. (2022) Scientific Reports, 12(1), 6034.
Religious coping, depression and anxiety among healthcare workers during the COVID-19 pandemic: A Malaysian perspective. (2021). Healthcare, 9 (1), 79. MDPI.
3. The international dimension of the sample is underdeveloped in the introduction. I would encourage the authors to clearly indicate why cross-cultural examination is relevant for understanding religiosity-health links and healthy aging.
4. For the method section, I would like the authors to clarify sampling procedure, including explicit inclusion/exclusion criteria. There is also no mention of validated instruments or scales for measuring health behaviors or psychological distress. Please clearly state how the questionnaire was developed or selected.
5. It is unclear how potential confounders (e.g., socioeconomic status, medical history beyond chronic illness) were controlled. The authors should explain explicitly how potential confounders were identified and managed or acknowledge this in the limitation.
6. The results are predominantly descriptive. Analytical interpretations linking lifestyle factors to outcomes are superficial. Please enhance results with deeper inferential analyses or multivariate approaches to examine associations between variables more robustly.
7. Some results sections provide ample detail (e.g., diet), while others (e.g., psychological distress and medication) lack clarity. I would encourage the authors to maintain consistent depth of detail across all result sections, particularly for psychological variables.
8. In the discussion, many findings are described without thorough psychological interpretation regarding underlying mechanisms or theoretical implications. There is a need to deepen the interpretation by discussing why observed patterns emerged, referencing theoretical frameworks mentioned earlier.
9. Certain claims, such as religiosity’s effect on health assessment and aging, are presented with undue certainty given the correlational design. The authors must temper claims with appropriate caution, clearly noting correlational limitations and avoiding causal language.
10. The writing style is often informal or overly narrative for a scientific article. Phrases like “we were fortunate to have conducted a research project…” or “it is of great interest…” introduce subjective tone. The authors should adopt a more objective and scholarly tone. Use precise, concise academic language.
11. The method section should be moved before results section
Author Response
This is an interesting study on life styles and health behaviorsamong consecrated women. I enjoyed reading the paper and appreciate the large sample size of this study. I have several comments to improve the manuscript further:
1. First, the introduction discusses the association between religion and health rather broadly, but it does not clearly articulate a specific theoretical framework. I would suggest the authors to clearly identify a psychological theory or model (e.g., Health Belief Model, Theory of Planned Behavior) to frame the researchThank you very much for this comment, indeed the introduction of the theory of planned behaviour will structure and help to better understand the results of our study.
2. Similarly, the introduction omits detailed exploration of potential mechanisms by which religiosity influences health. There are several potential mechanisms that can be discussed in the literature review. For instance, gratitude, often cultivated through religious practices like prayer and reflection, has been linked to improved emotional well-being and physical health. Social support within religious communities can buffer stress and reinforce health-promoting behaviors. Religious coping, such as viewing adversity through a spiritual lens, may help individuals manage stress more effectively and maintain psychological resilience. These should be highlighted to strengthen the introduction. Here are some papers that I think will be helpful and should be incorporated:
Social support and health: a review of physiological processes potentially underlying links to disease outcomes. (2006). Journal of behavioral medicine, 29, 377-387.
Dispositional gratitude, health-related factors, and lipid profiles in midlife: a biomarker study. (2022) Scientific Reports, 12(1), 6034.
Religious coping, depression and anxiety among healthcare workers during the COVID-19 pandemic: A Malaysian perspective. (2021). Healthcare, 9 (1), 79. MDPI.
This insightful remark has facilitated the expansion of knowledge in this domain and a more profound comprehension of the subject matter.
3. The international dimension of the sample is underdeveloped in the introduction. I would encourage the authors to clearly indicate why cross-cultural examination is relevant for understanding religiosity-health links and healthy aging.
This has been supplemented
4. For the method section, I would like the authors to clarify sampling procedure, including explicit inclusion/exclusion criteria. There is also no mention of validated instruments or scales for measuring health behaviors or psychological distress. Please clearly state how the questionnaire was developed or selected.
The inclusion criteria are women members of international apostolic religious congregations with papal privileges who wear religious garb. The inclusion criteria for the survey were to conduct a survey among nuns who have a similar vocation, who are not isolated by a cloister but wear a religious habit, who have international apostolic experience - this is possible thanks to subject to the general regulations issued by the Pope.
The study was designed as an anonymous survey with the objective of determining how members of religious communities assess their wellbeing in relation to community life and to identify potential avenues for improvement. The survey was disseminated via email in an electronic version (443) and in a paper version (20), depending on the participants' place of residence - Poland versus countries other than Poland. international, habitual, apostolic religious congregations in consultation with the Conference of Major Superiors of Religious Congregations in Poland. The data were collected over a two-month period, from April to June 2024. A preliminary study was conducted prior to the administration of the main survey, with the objective of validating the research tool.
5. It is unclear how potential confounders (e.g., socioeconomic status, medical history beyond chronic illness) were controlled. The authors should explain explicitly how potential confounders were identified and managed or acknowledge this in the limitation.
Thank you very much for drawing attention to this issue. Indeed, it is one of the limitations of the study involving recall error because for the most part the survey questionnaire relied on self-assessment of health behaviors.
The survey questionnaire is a proprietary tool developed on the basis of the research team's knowledge and experience. It was validated before the start of the main study and the results form the basis of a separate study.
6. The results are predominantly descriptive. Analytical interpretations linking lifestyle factors to outcomes are superficial. Please enhance results with deeper inferential analyses or multivariate approaches to examine associations between variables more robustly.
Our study related to many variables, we wanted to focus especially on those we considered important for the main objective, i.e. the reference to age and country of service, but of course we added more detailed analyses.
7. Some results sections provide ample detail (e.g., diet), while others (e.g., psychological distress and medication) lack clarity. I would encourage the authors to maintain consistent depth of detail across all result sections, particularly for psychological variables.
I would like to express my gratitude for your feedback. It is to be hoped that, following the implementation of the requested corrections, the results section will now be balanced.
8. In the discussion, many findings are described without thorough psychological interpretation regarding underlying mechanisms or theoretical implications. There is a need to deepen the interpretation by discussing why observed patterns emerged, referencing theoretical frameworks mentioned earlier.
Our work did not aim to analyse the psychological effects in detail, if only because the focus was mainly on self-assessment. Our aim was to examine the population of nuns in the context of health and we are aware that this is a broad topic that is difficult to analyse at all levels. We are very grateful for this comment, as the topic requires careful consideration and detailed research based on psychological tests in order to apply broader interpretations. Nevertheless, we have included references to the theory of planned behaviour and psychological ageing in the revised text.
9. Certain claims, such as religiosity’s effect on health assessment and aging, are presented with undue certainty given the correlational design. The authors must temper claims with appropriate caution, clearly noting correlational limitations and avoiding causal language.
This section has been edited
10. The writing style is often informal or overly narrative for a scientific article. Phrases like “we were fortunate to have conducted a research project…” or “it is of great interest…” introduce subjective tone. The authors should adopt a more objective and scholarly tone. Use precise, concise academic language.
This section has been edited
11. The method section should be moved before results section
This section has been edited
Reviewer 2 Report
Comments and Suggestions for Authors
The manuscript investigates the relationship between religiosity, lifestyle behaviors, and health outcomes in a sample of consecrated women. The topic is interesting and relevant; however, few issues need to be addressed to improve the clarity, and interpretation of your findings.
Major Comments
Data Interpretation and Statistical Reporting
Reliance on p-values:
The manuscript frequently emphasizes statistical significance (e.g., p < 0.05, p < 0.001) without reporting effect sizes or confidence intervals. Including these measures would provide a clearer picture of the practical significance of your findings.
Interpreting Medication Use vs. Chronic Illness:
The observation that the rate of medication use is approximately 20% higher than the self-reported incidence of chronic conditions is interpreted as a potential deficiency in disease awareness or care. However, alternative explanations—such as prophylactic prescriptions or treatment of subclinical conditions—should be discussed.
Subgroup Differences:
The abstract states that 57% of participants report leading a healthy lifestyle, yet subgroup analysis (e.g., 73% in the 65+ group) reveals marked differences. The manuscript should clarify these differences, provide a more nuanced interpretation, and discuss possible reasons for such variability.
Questionnaire and Measurement Issues
Definition and Validation of “Health Attitudes”:
The term “health attitudes” is used throughout, but the scale or composite measure underlying this construct is not clearly described. More details on the development, validation, and reliability of your survey instrument are necessary.
Operational Definition of “Healthy Lifestyle”:
It is unclear whether “leading a healthy lifestyle” is based on a composite score, a single self-report item, or a summation of several behaviors. A precise definition in both the Methods and Results sections would aid in the interpretation of your findings.
Study Design and Representativeness
Sample Imbalance:
Although the study purports an international scope, nearly 70% of the participants are Polish. This imbalance should be acknowledged and discussed in terms of how it might limit the generalizability of your results to other religious communities.
Cross-sectional Limitations:
As the study is cross-sectional, caution is needed when inferring directional relationships. Statements that suggest causality (e.g., that religiosity “leads to” better health outcomes) should be tempered to reflect that only associations were observed.
Overinterpretation and Causal Inferences
Some interpretations imply that religious practices or community life cause better health outcomes, yet the cross-sectional design only allows for correlation. It is recommended to reframe conclusions in associative terms and acknowledge potential confounding factors.
Minor Comments
Table 1: While demographic characteristics are clearly presented, consider clarifying how “No answer” responses were handled in your subsequent analyses.
Table 2: The grouping by “Poland,” “missionary,” and “non-missionary” countries needs clearer definitions. Explain the rationale and criteria for these categories in the Methods section.
Tables 3 and 4: The categorization of physical activity frequencies (e.g., “Every day,” “At least once a week”) is useful; however, it would help to standardize these categories and provide brief justification or reference for their use. Also, note the typographical errors (e.g., “Polan d Missionar y country” in Table 4) and ensure column headings are properly aligned.
Figures:
Ensure that all figures have clearly labeled axes and legends. In particular, Figures 3 and 5 include qualitative response categories that are not fully explained in the text. A concise legend or footnote should clarify how these qualitative categories were derived.
terms like “mean proportion” should be replaced with more standard statistical language.
Discussion of Limitations
Beyond noting the low availability of paper questionnaires, further discussion of potential biases (such as self-report bias, selection bias, and recall bias) is warranted. Additionally, the limitations inherent in a cross-sectional design should be more thoroughly discussed.
Best wishes
Author Response
The manuscript investigates the relationship between religiosity, lifestyle behaviors, and health outcomes in a sample of consecrated women. The topic is interesting and relevant; however, few issues need to be addressed to improve the clarity, and interpretation of your findings.
I would like to express my thanks to you for your thorough analysis of the manuscript and your comments.
Major Comments
Data Interpretation and Statistical Reporting
Reliance on p-values:
The manuscript frequently emphasizes statistical significance (e.g., p < 0.05, p < 0.001) without reporting effect sizes or confidence intervals. Including these measures would provide a clearer picture of the practical significance of your findings
Thank you for your feedback. The confidence interval used for the whole study is 95% at p<0.05 and 99.9% at p<0.001.
Interpreting Medication Use vs. Chronic Illness:
The observation that the rate of medication use is approximately 20% higher than the self-reported incidence of chronic conditions is interpreted as a potential deficiency in disease awareness or care. However, alternative explanations—such as prophylactic prescriptions or treatment of subclinical conditions—should be discussed.
We would like to express our gratitude for this intriguing suggestion, which has been incorporated into the manuscript.
Subgroup Differences:
The abstract states that 57% of participants report leading a healthy lifestyle, yet subgroup analysis (e.g., 73% in the 65+ group) reveals marked differences. The manuscript should clarify these differences, provide a more nuanced interpretation, and discuss possible reasons for such variability.
This remark is of significant value in the context of comprehending that such a favourable evaluation of one's lifestyle in the over-65 age group is derived from positive health habits, which are concomitant with psychological aspects. This phenomenon may be attributed to the accumulation of life experience, which is often accompanied by increased resilience.
Questionnaire and Measurement Issues
Definition and Validation of “Health Attitudes”:
The term “health attitudes” is used throughout, but the scale or composite measure underlying this construct is not clearly described. More details on the development, validation, and reliability of your survey instrument are necessary.
The survey questionnaire is a proprietary tool developed on the basis of the research team's knowledge and experience. It was validated before the start of the main study and the results form the basis of a separate study
Operational Definition of “Healthy Lifestyle”:
It is unclear whether “leading a healthy lifestyle” is based on a composite score, a single self-report item, or a summation of several behaviors. A precise definition in both the Methods and Results sections would aid in the interpretation of your findings.
The term 'living a healthy lifestyle' is a single element of self-assessment, although it has a statistical correlation with both the proposed description of health behavior and lifestyle elements such as physical activity.
Study Design and Representativeness
Sample Imbalance:
Although the study purports an international scope, nearly 70% of the participants are Polish. This imbalance should be acknowledged and discussed in terms of how it might limit the generalizability of your results to other religious communities.
This is included as a limitation of the study
Cross-sectional Limitations:
As the study is cross-sectional, caution is needed when inferring directional relationships. Statements that suggest causality (e.g., that religiosity “leads to” better health outcomes) should be tempered to reflect that only associations were observed.
This comment has been included in the text.
Overinterpretation and Causal Inferences
Some interpretations imply that religious practices or community life cause better health outcomes, yet the cross-sectional design only allows for correlation. It is recommended to reframe conclusions in associative terms and acknowledge potential confounding factors.
Expressions of gratitude are extended for highlighting this issue. Indeed, it constitutes one of the study's limitations, as recall error can act as a confounding factor. This is particularly salient given the survey questionnaire's reliance on self-reported health behaviors. This limitation is acknowledged in the study's section on limitations.
Minor Comments
Table 1: While demographic characteristics are clearly presented, consider clarifying how “No answer” responses were handled in your subsequent analyses.
‘No answer’ has been included in subsequent statistical analyses.
Table 2: The grouping by “Poland,” “missionary,” and “non-missionary” countries needs clearer definitions. Explain the rationale and criteria for these categories in the Methods section.
This point is clarified in the ‘method’
Tables 3 and 4: The categorization of physical activity frequencies (e.g., “Every day,” “At least once a week”) is useful; however, it would help to standardize these categories and provide brief justification or reference for their use. Also, note the typographical errors (e.g., “Polan d Missionar y country” in Table 4) and ensure column headings are properly aligned.
The categorisation of physical activity frequency was the result of the validation process, during which the participants of the preliminary study indicated in their comments the necessity to expand the criteria.
Figures:
Ensure that all figures have clearly labeled axes and legends. In particular, Figures 3 and 5 include qualitative response categories that are not fully explained in the text. A concise legend or footnote should clarify how these qualitative categories were derived.
terms like “mean proportion” should be replaced with more standard statistical language.
The text passages affected by the aforementioned comments have been corrected.
Discussion of Limitations
Beyond noting the low availability of paper questionnaires, further discussion of potential biases (such as self-report bias, selection bias, and recall bias) is warranted. Additionally, the limitations inherent in a cross-sectional design should be more thoroughly discussed.
The text passages affected by the aforementioned comments have been corrected.
Round 2
Reviewer 1 Report
Comments and Suggestions for Authors
I appreciate the authors' efforts in revising and addressing my comments. However, some of the comments were not addressed. Below, I will describe some of my previous comments that were not addressed well:
1. It is unclear how potential confounders (e.g., socioeconomic status, medical history beyond chronic illness) were controlled. The authors should explain explicitly how potential confounders were identified and managed or acknowledge this in the limitation.
For this comment, the authors addressed by acknowledging recall bias. But this comment is more about the possibility of confounders. This was not acknowledged in the limitation.
2. Similarly, the introduction omits detailed exploration of potential mechanisms by which religiosity influences health. There are several potential mechanisms that can be discussed in the literature review. For instance, gratitude, often cultivated through religious practices like prayer and reflection, has been linked to improved emotional well-being and physical health. Social support within religious communities can buffer stress and reinforce health-promoting behaviors. Religious coping, such as viewing adversity through a spiritual lens, may help individuals manage stress more effectively and maintain psychological resilience. These should be highlighted to strengthen the introduction. Here are some papers that I think will be helpful and should be incorporated:
Social support and health: a review of physiological processes potentially underlying links to disease outcomes. (2006). Journal of behavioral medicine, 29, 377-387.
Dispositional gratitude, health-related factors, and lipid profiles in midlife: a biomarker study. (2022) Scientific Reports, 12(1), 6034.
Religious coping, depression and anxiety among healthcare workers during the COVID-19 pandemic: A Malaysian perspective. (2021). Healthcare, 9 (1), 79. MDPI.
For this comment, the revised paragraph should start with a topic sentence on potential mechanisms by which religiosity influences health. Some of the newly included citations were not particularly relevant and the suggested citations in the previous review were not incorporated. The authors should include more citations to further support the points made.
3. For the method section, I would like the authors to clarify sampling procedure, including explicit inclusion/exclusion criteria. There is also no mention of validated instruments or scales for measuring health behaviors or psychological distress. Please clearly state how the questionnaire was developed or selected.
For this comment, there is no mention on the questionnaires and scales used for the current study. The sampling procedure was described vaguely too.
4. The writing of the papers can be improved further to meet the standard for peer-reviewed publication. I have detailed my comments regarding the writing quality in "Comments on the Quality of English Language".
Comments on the Quality of English Language1. Numerous paragraphs begin abruptly without a clear topic sentence or logical transition from the preceding paragraph. In the Introduction and Discussion sections, new ideas are often introduced without framing or linking to the previous content. Please revise each paragraph to include a clear topic sentence that previews its content. Use logical connectors (e.g., "In contrast," "Moreover," "Consequently") to improve flow.
2. There are many paragraphs consisting of only one sentence, particularly in the Methods, Results, and Discussion sections. Please combine related single-sentence paragraphs where appropriate. Group sentences that elaborate on the same finding or idea into cohesive paragraphs. Some sections (especially Discussion and Results) include extremely long paragraphs covering multiple ideas, while others are underdeveloped.
3. Several sentences are overly long or verbose without adding new conceptual content.
4. The Results section lacks clear subheadings that reflect a coherent structure (e.g., Health Behaviors, Psychological Well-being, Attitudes).
5. The abstract is too long and includes excessive detail, making it harder to extract the key findings. Please focus the abstract on study aims, central findings, and broader implications.
6. The manuscript contains awkward sentence constructions and inconsistent verb tenses (e.g., switching between past and present).
Author Response
I appreciate the authors' efforts in revising and addressing my comments. However, some of the comments were not addressed. Below, I will describe some of my previous comments that were not addressed well:
- It is unclear how potential confounders (e.g., socioeconomic status, medical history beyond chronic illness) were controlled. The authors should explain explicitly how potential confounders were identified and managed or acknowledge this in the limitation.
For this comment, the authors addressed by acknowledging recall bias. But this comment is more about the possibility of confounders. This was not acknowledged in the limitation.
We did not include tools for controlling disruptive factors in our study. It was included as part of the limitation
- Similarly, the introduction omits detailed exploration of potential mechanisms by which religiosity influences health. There are several potential mechanisms that can be discussed in the literature review. For instance, gratitude, often cultivated through religious practices like prayer and reflection, has been linked to improved emotional well-being and physical health. Social support within religious communities can buffer stress and reinforce health-promoting behaviors. Religious coping, such as viewing adversity through a spiritual lens, may help individuals manage stress more effectively and maintain psychological resilience. These should be highlighted to strengthen the introduction. Here are some papers that I think will be helpful and should be incorporated:
Social support and health: a review of physiological processes potentially underlying links to disease outcomes. (2006). Journal of behavioral medicine, 29, 377-387.
Dispositional gratitude, health-related factors, and lipid profiles in midlife: a biomarker study. (2022) Scientific Reports, 12(1), 6034.
Religious coping, depression and anxiety among healthcare workers during the COVID-19 pandemic: A Malaysian perspective. (2021). Healthcare, 9 (1), 79. MDPI.
For this comment, the revised paragraph should start with a topic sentence on potential mechanisms by which religiosity influences health. Some of the newly included citations were not particularly relevant and the suggested citations in the previous review were not incorporated. The authors should include more citations to further support the points made.
I would like to express my gratitude for your suggestion. The sources mentioned were used to strengthen the analysis of the mechanisms of the impact of religion on health.
- For the method section, I would like the authors to clarify sampling procedure, including explicit inclusion/exclusion criteria. There is also no mention of validated instruments or scales for measuring health behaviors or psychological distress. Please clearly state how the questionnaire was developed or selected.
For this comment, there is no mention on the questionnaires and scales used for the current study. The sampling procedure was described vaguely too.
The questionnaire employed in this study is an original instrument developed by the research team. This instrument was developed by the research team, who had extensive clinical experience and had conducted a comprehensive review of the relevant literature. To ensure the validity and reliability of the instrument, a pilot study was conducted to assess the questions' clarity, unambiguity, and content adequacy. Following the analysis of the pilot study results, the research team made the necessary modifications to the questionnaire, thus ensuring its validation prior to its implementation in the main study.
Inclusion and exclusion criteria are included in the text.
- The writing of the papers can be improved further to meet the standard for peer-reviewed publication. I have detailed my comments regarding the writing quality in "Comments on the Quality of English Language".
Comments on the Quality of English Language
- Numerous paragraphs begin abruptly without a clear topic sentence or logical transition from the preceding paragraph. In the Introductionand Discussionsections, new ideas are often introduced without framing or linking to the previous content. Please revise each paragraph to include a clear topic sentence that previews its content. Use logical connectors (e.g., "In contrast," "Moreover," "Consequently") to improve flow.
- There are many paragraphs consisting of only one sentence, particularly in the Methods, Results, and Discussion sections. Please combine related single-sentence paragraphs where appropriate. Group sentences that elaborate on the same finding or idea into cohesive paragraphs. Some sections (especially Discussion and Results) include extremely long paragraphs covering multiple ideas, while others are underdeveloped.
- Several sentences are overly long or verbose without adding new conceptual content.
- The Results section lacks clear subheadings that reflect a coherent structure (e.g., Health Behaviors, Psychological Well-being, Attitudes).
- The abstract is too long and includes excessive detail, making it harder to extract the key findings. Please focus the abstract on study aims, central findings, and broader implications.
- The manuscript contains awkward sentence constructions and inconsistent verb tenses (e.g., switching between past and present).
Reviewer 2 Report
Comments and Suggestions for Authors
Thank you for addressing the comments
Author Response
Thank you for addressing the comments.
Many thanks for your review